# Antibiotic-Induced Immunosuppression—A Focus on Cellular Immunity

**DOI:** 10.3390/antibiotics13111034

**Published:** 2024-11-01

**Authors:** Timothy Arthur Chandos Snow, Mervyn Singer, Nishkantha Arulkumaran

**Affiliations:** Bloomsbury Institute of Intensive Care Medicine, University College London, London WC1E 6DH, UK; timothy.snow@doctors.net.uk (T.A.C.S.); m.singer@ucl.ac.uk (M.S.)

**Keywords:** antimicrobial, critical illness, immune response, infection, lymphocyte, monocyte, sepsis

## Abstract

Antibiotics are the fundamental treatment for bacterial infections. However, they are associated with numerous side effects. Their adverse effects on the immune system are increasingly recognised, with several mechanisms identified. In this review, we focus on their direct effects on cellular immunity. We review the effects of antibiotics on mitochondrial function and how they impair specific immune cell functions including chemotaxis, phagocytosis, cytokine production, antigen presentation, and lymphocyte proliferation. Findings are described in a multitude of *in vivo* and *in vitro* models. However, their impact on patient immunity and clinical outcomes requires further research. Awareness of the potential adverse effects of antibiotics may improve antimicrobial stewardship. The use of therapeutic drug monitoring may help to reduce dose-dependent effects, which warrants further research.

## 1. Introduction

Antibiotics are the fundamental treatment for bacterial infections. However, given the lack of suitable rapid diagnostic tests, most patients with sepsis are commenced on broad-spectrum antibiotics and transitioned to narrow-spectrum if cultures identify a causative organism. Given the poor sensitivity of traditional cultures, many patients are not de-escalated to narrow-spectrum antibiotics. Additionally, the length of antibiotic course is highly variable, often between 3 and 14 days [1].

Many patients with sepsis develop immunosuppression, increasing their risk of secondary infection [2]. Mechanisms underpinning sepsis-induced immunosuppression are multifactorial but likely to include off-target effects of medications, including antibiotics. Although adverse effects of antibiotics on immune cell function are well described [3,4], their specific effects on immunosuppression after sepsis and critical illness are unknown. Prolonged use of antibiotics may exacerbate this immunosuppression, leaving septic patients more vulnerable to subsequent infections [5].

Data on antibiotic modulation of immunity have been mainly characterised in cell lines and animal models [4,6]; clinical data are limited [7]. Most antibiotic classes suppress both innate and adaptive immune responses [6]. It is imperative to understand off-target immune effects of specific antibiotic classes and to determine underlying mechanisms.

Antibiotics target (prokaryotic) bacterial cellular processes, although the antibiotic-related side effects experienced by patients clearly indicate off-target effects [8]. It is unclear if the mechanism(s) by which antibiotics impact on human immune cells are directly related to their antibacterial effects on DNA (deoxyribonucleic acid) transcription (ciprofloxacin) or protein translation (clarithromycin, gentamicin). The effect of beta-lactams on human immune cells is clearly unrelated to their mechanism of action on bacteria. Several pathways have been implicated in antibiotic-induced immunosuppression (Figure 1 and Appendix A).

In this review, we describe the effects of antibiotics on effects on mitochondrial function and specific immune cell functions including chemotaxis, phagocytosis, cytokine production by granulocytes and monocytes, antigen presentation (monocytes), and lymphocyte apoptosis and proliferation. Findings are described in a multitude of *in vivo* and *in vitro* models which highlight the potential relevance of the findings to clinical practise.

## 2. Cellular Dysfunction

### 2.1. Mitochondrial Dysfunction

Mitochondria are integral to regulating immune function; defects in leukocyte energy metabolism in septic patients are associated with immunosuppression [9]. The direct roles of mitochondria in innate and adaptive immune cells are wide-ranging, suggesting that mitochondrial dysfunction may play a significant causative role [10]. Given the current understanding of the prokaryotic origins of mitochondria, it is plausible that antibiotics targeting bacteria has detrimental effects on mitochondrial functionality.

For example, electron transport chain (ETC) adaptations serve as an early immunological–metabolic checkpoint for innate immune responses to bacterial infection [11]. Synthesis of mitochondrial DNA induced after the engagement of Toll-like receptors (TLRs) mediates NLRP3 (NOD-, LRR-, and pyrin domain-containing protein 3) inflammasome signalling in macrophages [12]. Antibiotics including lincosamides, macrolides, and fluoroquinolones accumulate in phagocytes and may interfere with the above processes [13]. The highly energy-dependent respiratory burst required for bacterial killing by macrophages is impaired by a dose-dependent inhibition of mitochondrial respiratory activity by ciprofloxacin [14].

The effects of antibiotics on immune system function are complex; observations from *in vitro* experiments may not necessarily translate to the *in vivo* situation. For instance, ciprofloxacin decreases the release of IL-1ß from human volunteer monocytes stimulated for 24 h in vitro with lipopolysaccharide (LPS) [15]; however, after administration of oral ciprofloxacin to healthy volunteers for 7 days, *in vitro* LPS stimulation enhanced IL-1ß production [16]. Previous work by our group has also demonstrated that ciprofloxacin and LPS *in vitro* stimulation suppresses pro-inflammatory cytokine release from volunteer and septic patient PBMCs after 24 h but not via mitochondrial pathways [5]. Whether these conflicting findings are due to differences between *in vivo* or *in vitro* models or due to the differences in duration of antibiotic administration remains to be elucidated. Animal models or cells isolated from septic patients administered ciprofloxacin could be used to confirm these findings.

Aminoglycoside antibiotics are a family of amino-modified sugars containing hydrophilic portions and cationic amine moieties that preferentially bind nucleic acids due to their negative charge. They can cause translational errors and the assembly of incorrect amino acid products or premature termination of protein synthesis [17,18]. While their effects on immune cell mitochondria are yet to be delineated, they do impact upon renal tubular epithelial mitochondria in animal models [19].

Aminoglycosides bind to human mitochondrial ribosomes [20]. In isolated mitochondria from rat renal tubular cells, aminoglycosides induced ETC uncoupling, increased mitochondrial membrane cation permeability [19], and collapse of the mitochondrial membrane potential [21]. This reduced oxidative phosphorylation [22] and the production of mitochondrial reactive oxygen species (ROS) [23]. However, there may be differing effects on different aspects of mitochondrial respiration [24], which may be why some studies demonstrated an increase in ROS [25]. Oxazolidinone antibiotics bind to mitochondrial ribosomes, reducing mitochondrial protein in non-immune cells [26] and the K562 lymphoblastoid and THP-1 monocyte cell lines [27,28].

In a rat model of gentamicin-induced renal toxicity, respiratory components including cytochrome C and NADH (nicotinamide adenine dinucleotide (NAD) + hydrogen (H)) were depleted. This was associated with an opening of the mitochondrial transition pore and an increase in ROS production [29]. The potency of the aminoglycosides in producing these effects correlates with the number of ionizable amino groups present on the aminoglycoside molecule, suggesting that cationic charge is an important molecular determinant of toxic effect [24]. Similar effects have been demonstrated in other cell types including mouse cochlear cells [30] but not liver cells [22], suggesting certain cell types are at increased risk. Mitochondria in peripheral blood monocular cells (PBMCs) may not be affected [5]; however, further research is required to explore the effects of aminoglycosides on mitochondrial function in other immune cell types.

### 2.2. Chemotaxis and Migration

Immune cells migrate from the blood to the source of infection via a chemokine gradient.

Mouse macrophage chemotaxis was increased by carbapenems [31] and by teicoplanin and vancomycin [32] but decreased by amoxicillin beta-lactams, clindamycin, and tetracycline [33]. Mouse neutrophil migration was decreased by linezolid [34], and rat neutrophil migration was increased by colistin [35].

In volunteer immune cells and PBMCs, erythromycin and roxithromycin increased migration or chemotaxis [36], while aminoglycosides and tetracyclines were inhibitory [37,38]. Penicillins [38,39], carbapenems [39,40], and linezolid had no effect [41,42]. Cephalosporins [37,39,43,44,45,46,47], teicoplanin [48,49], and vancomycin had differing model-dependent effects [49,50]. In *in vivo* healthy volunteer models, erythromycin impaired neutrophil migration via reduced IL-8 [51], and ceftriaxone impaired chemotaxis [52].

In patients, macrolides inhibited neutrophil chemotaxis and migration predominantly through reduced IL-8 in patients with COPD [53,54,55,56,57], bronchial hyperreactivity [58], chronic sinusitis [59,60,61,62], and allergy [63]. This effect was, however, not seen consistently with clarithromycin, although this be related to the different diseases studied [64].

### 2.3. Toll-Like Receptor Expression

Toll-like receptors (TLRs) are pattern recognition receptors which are expressed on a variety of immune cells including neutrophils and monocytes. TLRs are activated by numerous bacterial products including peptidoglycans and LPS. Activation is important in the initiation of the inflammatory cascade in response to infection [65].

In a mouse model, folimycin decreased surface expression of TLR mediated by inhibition of V-ATPases (vacuolar adenosine triphosphate-ases) [66]. In THP-1 cell lines, linezolid increased TLR expression (-1, -2, and -6), while daptomycin decreased it [67]. Erythromycin, moxifloxacin, and doxycycline increased TLR expression (-1, -2, -4, and -6) both in the THP-1 cell line and in patients following cardiac bypass [68]. These findings need to be confirmed in other diseases.

### 2.4. Cytokine Release

Cytokines are inflammatory mediators released by several cell types in response to infection which regulate the inflammatory and immune response. They are broadly characterised into pro-inflammatory (e.g., IL-1β and TNF-α), anti-inflammatory (e.g., IL-10), or mixed (IL-6).

Most antibiotics inhibit cytokine production and release. In *ex vivo* mouse models on antigen-presenting cells, roxifloxacin [69], erythromycin [70], azithromycin [70], and doxycycline inhibited the release of multiple pro-inflammatory cytokines [71]. One postulated mechanism was through the inhibition of mitochondrial protein translation and NLRP3 inflammasome assembly in bone marrow-derived macrophages [71].

Using *in vivo* and *in vitro* animal models, fluroquinolones inhibited some pro-inflammatory cytokines, although there were in-class differences related to antibiotic structure [72,73]. Macrolides were anti-inflammatory [74], while roxifloxacin had time-dependent effects, increasing pro-inflammatory release initially but causing inhibition after over 2 weeks’ treatment [75,76]. Linezolid and vancomycin also reduced cytokine release in pneumonia models [34,77,78,79,80]. In large animal pneumonia models, azithromycin inhibited IL-6 release [81], linezolid had no effect [82], and danofloxacin was predominately anti-inflammatory, reducing pro-inflammatory cytokine release yet increasing IL-10 [83].

In J774 macrophage cell lines, macrolides inhibited pro-inflammatory cytokine release through reduced COX-2 (cyclooxygenase-2) and nitric oxide synthase (NOS) expression [74]. In THP-1 monocyte cell lines, linezolid and vancomycin increased both pro- and anti-inflammatory cytokine release [67]; erythromycin, doxycycline, and moxifloxacin increased pro-inflammatory cytokine release [68]; grepafloxacin inhibited pro-inflammatory release [84]; while daptomycin had mixed effects on pro-inflammatory cytokine release [67].

In volunteer whole blood and PBMC models, cytokine release was reduced by linezolid [85,86], clindamycin [87], teicoplanin [88], erythromycin [86,89], ceftazidime [90], and tigecycline [91]. Meropenem had mixed effects, reducing the release of some pro-inflammatory cytokines [40]. Amoxicillin and trimethoprim, however, were pro-inflammatory [91,92], while penicillin and metronidazole had no effect [87,89,93]. Several studies yielded conflicting results. Vancomycin either decreased release or had no effect [85,93], while fluroquinolones either reduced [15,40,90,94,95,96,97,98,99,100], had no effect [101], or increased release [102].

In patient studies, clarithromycin given to COPD (chronic obstructive pulmonary disease) and asthmatic patients either had no effect [57,103] or reduced both pro- and anti-inflammatory cytokine release [104,105]. Erythromycin given to wheezy children reduced cytokine levels [106], while amoxicillin and penicillin given to allergy patients increased pro-inflammatory cytokine levels [107]. Norfloxacin in cirrhotic patients induced an immunosuppressive phenotype with an increased proportion of T_regs_ and IL-10 release [108]. Suggested mechanisms include a direct fluroquinolone effect on protein synthesis [15,94,99], mitochondrial ETC inhibition [100], inhibition of COX-2 [98], and upregulation of the *rag1* (recombination activating gene 1) gene (responsible for T-cell receptor formation) [108].

Clarithromycin has different effects dependant on the patient population studied. In those with ventilator-associated or community-acquired pneumonia, it causes an increase in LPS-induced TNF-α release [109,110]. However, in patients with septic shock, it reduced release, but IL-6 release was maintained [110]. This suggests there may be a disease- or illness-severity-specific effect, although whilst the stimulation studies were performed at day 4 of enrolment, it was unclear whether there were differences in duration of admission prior to enrolment, which could also explain the findings. However, the different effects of clarithromycin on cytokine release highlights potential mechanisms to be explored.

### 2.5. Phagocytosis

Phagocytosis is an important early antibacterial mechanism of professional phagocytotic cells, including neutrophils and monocytes, for the elimination of pathogens. Antigen-presenting cells, including monocytes, present antigens processed from phagocytosed bacteria to trigger the adaptive immune system [111].

In mouse and rat macrophages, carbapenems increased phagocytosis [31], while amoxicillin, clindamycin, azithromycin, and erythromycin impaired it [33,70]. Vancomycin and teicoplanin had differing effects with both enhancement and impairment [32,112]. Daptomycin and lomefloxacin had no effect [112,113]. In the THP-1 cell line, antifungal agents suppressed phagocytosis [114].

In volunteer immune cells and PBMCs, meropenem and macrolides reduced neutrophil phagocytosis [40,115]. Cephalosporins, co-amoxiclav, and imipenem increased neutrophil phagocytosis [44,45,46,115,116,117], while macrolides increased monocyte phagocytosis [118]. Rokitamycin and linezolid had no effect [41,42,119]. Teicoplanin and vancomycin had differing dose-dependent effects [48,49]. Fluroquinolones including ciprofloxacin also had differing effects, with low doses enhancing phagocytosis [120,121] or having no effect [113,122], while inhibition could occur at supra-pharmacological doses [123].

In volunteer *in vivo* models, carbapenems increased phagocytosis [117], while ceftriaxone had no effect [52]. In patients, piperacillin, doxycycline, and moxifloxacin inhibited monocyte phagocytosis after cardiac surgery [68]. Azithromycin increased macrophage phagocytosis [124], and clarithromycin increased neutrophil phagocytosis in COPD patients [64]. Roxifloxacin also increased neutrophil phagocytosis [125].

Given the myriad of conflicting results with individual antibiotics or classes, likely related to different experimental conditions and the cell types studied, *in vivo* antibiotic dosing models would be required to answer definitively even if most phagocytosis assays are *in vitro*.

### 2.6. Antigen Presentation

Antigen presenting cells are key mediators between the innate and adaptive immune systems. Phagocytosed bacterial products are broken down into antigens and presented on the cell surface by major histocompatibility complex class II (MHC class II) molecules including Human Leukocyte Antigen—DR isotype (HLA-DR) to the T-cell receptor on lymphocytes. The response of the lymphocyte to the selected antigen is regulated by co-stimulatory molecules, of which monocyte CD80 and CD86 are well characterised [126]. Impaired antigen presentation is a common feature of sepsis-induced immunosuppression and is associated with the development of secondary infections and mortality [127,128].

In mice, roxithromycin impairs antigen-presenting cell MHC class II presentation [69] and CD80 and CD86 on B-cells [129,130], although this effect was only seen with longer courses [131].

In volunteer PBMCs, pefloxacin and ciprofloxacin had no effect on antigen presentation [132]. In PBMCs isolated from patients with allergies, there was an upregulation of HLA-DR, CD80, and CD86 with amoxicillin [133], while in PBMCs from cirrhotic patients, norfloxacin impaired CD80/86 expression [108]. It is unclear whether the different effect of norfloxacin compared to ciprofloxacin is related to the drug itself or the effect of cirrhosis in this population. Macrolides increased CD80 but not HLA-DR in patients with chronic sinusitis [134], while clarithromycin increased HLA-DR in patients with pneumonia and sepsis [135] and increased CD86 in patients with ventilator-associated pneumonia and sepsis [109]. The differences in these findings appear to be disease-related, with greater effect seen in acute illness with short duration of antibiotics compared to chronic illness with prolonged antibiotic course. A prolonged time course of changes in antigen presentation with clarithromycin would be required to confirm this, especially given changes between the clarithromycin and placebo groups were only seen 10 days after treatment [135].

### 2.7. Lymphocyte Proliferation

In response to infection, lymphocytes undergo clonal proliferation and differentiation into the various subclasses to facilitate bacterial clearance. Imbalances in proliferation have been associated with mortality in sepsis with reduced proliferation of T_helper_ cells but increased proliferation of the immunosuppressive T_reg_ type [136].

Antibiotics have been demonstrated to inhibit cell proliferation. In volunteer PBMCs, fluroquinolones impair proliferation through alterations in IL-2 release and increased monocyte prostaglandin E2 release [95,98,102]. Additional mechanistic work in breast and lung cancer cell lines suggests the effect could also be caused by damaging mitochondria and deactivating the PI3K/Akt/mTOR (phosphatidylinositol 3-kinase/Ak strain transforming/mechanistic target of rapamycin) and MAPK/ERK (mitogen-activated protein kinase/extracellular signal-regulated kinases) pathways [137,138]. Another potential mechanism could be through direct binding and inhibition of the T-cell receptor. In patients with allergies to fluroquinolones and amoxicillin, these antibiotics directly bind to the T-cell receptor, albeit stimulating proliferation in these cases [133,139].

Erythromycin, clindamycin, rifampicin, fusidic acid, nitrofurantoin, and doxycycline all inhibited proliferation of healthy volunteer lymphocytes, whereas penicillin, cephalosporins, aminoglycosides, chloramphenicol, sulfamethoxazole, trimethoprim [140], and macrolides did not [141,142]. However, in a mouse model, cefotaxime did inhibit lymphocyte proliferation [143]. Other cephalosporins and penicillins (including piperacillin) impaired proliferation; however, the effect of each antibiotic was not consistent amongst different cell types when comparing their effect on proliferation of chick embryos, lymphocyte cell lines, and mouse lymphocytes. This cell-specific effect may in part explain the conflicting results described; further research is required to identify the causative mechanism and whether the *in vitro* effect is seen *in vivo* [144].

### 2.8. Lymphocyte Apoptosis

Following clonal lymphocyte proliferation after infection, apoptosis occurs, leaving the memory cells quiescent, primed for subsequent re-infections. Sepsis is associated with increased apoptosis, causing lymphopenia and increased risk of subsequent infection [128].

Lymphocyte apoptosis is mediated by two main pathways, mitochondrial (which includes caspases-3 and -8 and Bcl-2 (B-cell lymphoma) proteins) and non-mitochondrial pathways [145].

Linezolid induced lymphocyte apoptosis through mitochondrial pathways by inhibiting mitochondrial protein synthesis and complex IV activity in volunteer PBMCs and skin nerve fibres [146] and in patient and rat skeletal muscle and liver [147]. Protein levels were reduced, while mitochondrial DNA levels remained similar, suggesting a direct action on the mitochondrial ribosome; certain polymorphisms appear to be at increased risk. Moxifloxacin increased murine macrophage cell death, although this could be ameliorated by the use of immunomodulatory compounds tinrostim and licopid [148].

The experimental beta-lactam, lactam 1, induced T-cell apoptosis in a Jurkat cell line [149] and a mouse breast cancer model [150] through direct damage to, and inhibition of, DNA replication. This led to p38 MAPK activation, S phase arrest, and apoptotic cell death mediated by caspase-3, -8, and -9 activation, cleavage of the pro-apoptotic Bcl-2 family protein Bid (BH3-interacting domain death agonist), and release of mitochondrial cytochrome c.

The fluroquinolone ciprofloxacin also induced Jurkat cell apoptosis through mitochondrial pathways by causing direct damage to mitochondrial DNA, inhibiting the respiratory chain and decreasing membrane potential [151]. Similar effects of mitochondrial-induced apoptosis have been demonstrated by ciprofloxacin on other cell lines, including colon and bladder tumour cells [152,153], and by levofloxacin in breast and lung cancer cell lines [137,138]. Whilst this suggests a consistent mechanism, the effect on primary lymphocytes and other immune cell types remains to be delineated.

Gentamicin-induced ETC inhibition activated caspases-3 and -9, leading to mitochondrial-induced cellular apoptosis in renal cell lines [154,155].

## 3. Clinical Consequences

Antibiotics remain the key management strategy for the treatment of bacterial infection. In light of the adverse effects on immune cell function described in this review, and the multitude of other deleterious effects on antimicrobial resistance and on the gut microbiota [6], we advocate strong antimicrobial stewardship [156], key features of which include the following:

Judicious initiation of antibiotics—The diagnosis of infection is not always straightforward. Recognising this, recent guidance now recommends consideration of illness severity as a guide to the urgency of antibiotic administration [157]. This allows time for the collection of appropriate cultures and consideration to many of the infection mimics prior to commencing antibiotics, especially as early antibiotic discontinuation even if infection is disproved is not always performed [158].

Reduced inappropriate usage—When antibiotics are required, broad-spectrum antibiotics are sometimes initiated, or the courses of antibiotics are sometimes unduly prolonged as a precaution. This is especially pertinent for community-acquired infections and surgical prophylaxis [159]. However, (appropriately) shorter duration of antibiotic courses are non-inferior to longer courses and may be associated with lower risk of development of subsequent infections [160,161].

Selecting narrow- over broad-spectrum antibiotics—Overuse of broad-spectrum antibiotics when not required is associated with increased risk of antimicrobial resistance but may also be associated with an increased risk of mortality [162]. Our review suggests carbapenems and piperacillin may have greater immunosuppressive effects over cephalosporins and amoxicillin—however, this needs to be explored further.

Therapeutic drug monitoring (TDM)—Many of the immunosuppressive effects demonstrated in this review were identified at higher concentrations. Whilst most antibiotic dosing is based on pharmacokinetic/pharmacodynamic studies in healthy volunteers, measured serum concentrations in septic patients vary significantly with both relative under- and overdosing demonstrated [163]. With the increasing interest in TDM, antibiotic dosing could be optimised to minimise the risk of supra-clinical concentrations which are associated with increased 28-day mortality potentially mediated through their deleterious immunosuppressive and other adverse effects [164].

## 4. Summary

Antibiotics are associated with multiple deleterious effects beyond their immediate side-effect profile. These include patient-specific effects of idiosyncratic drug reactions, disruption of microbiome and mitochondrial toxicity, and population-level effects including antimicrobial resistance. A growing body of evidence shows antibiotics can directly impact immune cell function, although their extent and the mechanisms by which these occur remains relatively unexplored.

Critical illness is associated with multiple immune defects which are associated with an increased risk of subsequent infections. While reduced monocyte HLA-DR expression and lymphopenia are well described, it is unclear whether these are isolated defects or symptomatic of wider immune cell dysfunction. The lack of benefit demonstrated by immunomodulatory treatments targeting these pathways suggests the latter; however, more research is required to explore this further.

Given the significant use of antibiotics in the critically ill, it is plausible that antibiotics may directly affect immune cell function, exacerbating the immunosuppressive state seen in critical illness. Confirmation of this would add support to rigorous antimicrobial stewardship goals aiming to reduce undue antimicrobial use, especially if the deleterious effects are related to duration of course or use of broad-spectrum agents or if there is evidence of a dose-dependent effect.

Given beta-lactam antibiotics are the most widely used class of antibiotics in the critically ill, they represent the best target for identification of immunosuppressive effects. The growing use of therapeutic drug monitoring for them also presents an opportunity to incorporate dosing regimens which ensure appropriate serum concentrations for bacterial killing whilst preventing supra-clinical concentrations which could have deleterious effects on immune cell function.

## Figures and Tables

**Figure 1 antibiotics-13-01034-f001:**
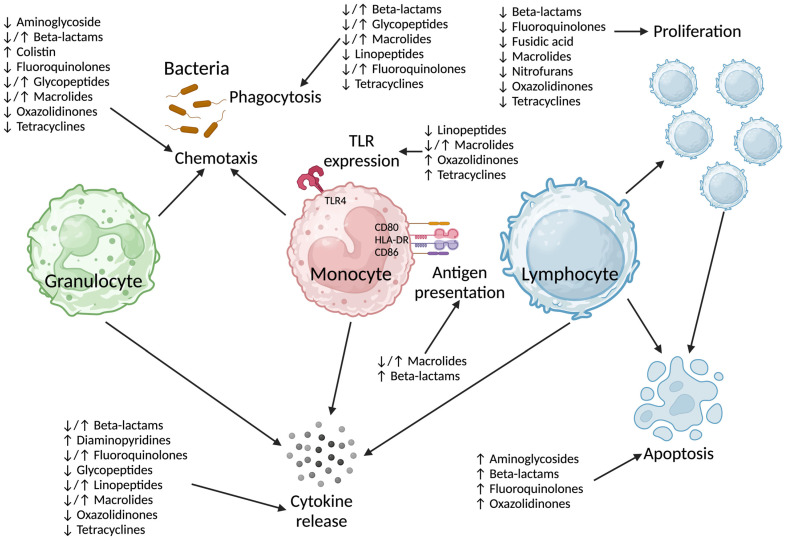
Summary of immunosuppressive effects of antibiotics by class on immune cell function. TLR: Toll-like receptor, HLA-DR: Human Leukocyte Antigen—DR isotype, CD: cluster of differentiation. Created in www.BioRender.com.

## Data Availability

The original contributions presented in the study are included in the article/Appendix A; further inquiries can be directed to the corresponding author.

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
