# Peer review of "Antibiotic-Induced Immunosuppression—A Focus on Cellular Immunity"

_antibiotics, 2024, doi:10.3390/antibiotics13111034_

Round 1

Reviewer 1 Report

Comments and Suggestions for Authors

This manuscript is excellent addition to the filed. The authors have comprehensively compiled the studies that have advanced this filed. Though there are minor language issues with this manuscript that need to be revised. I recommend this article for the publication.

Comments on the Quality of English Language

English revision is required.

Author Response

This manuscript is excellent addition to the filed. The authors have comprehensively compiled the studies that have advanced this filed. Though there are minor language issues with this manuscript that need to be revised. I recommend this article for the publication.

Thank you for your comments

Reviewer 2 Report

Comments and Suggestions for Authors

The authors addressed an increasingly important negative side effects of antibiotics on immunity despite the beneficial effect in treatment and fighting infections, especially bacterial ones.

However, apart from the recognized and appreciated narration of the published studies with focus on the context of sepsis and the negative effect of antibiotics on cellular immunity, the manuscript is, in some parts, lacking the appropriate critical review on the published data and the clear complete drafting of the cited results  e.g., when the results of several studies on the same point are different or contradictory, the review drafting authors should critically discuss the possible reason of difference, if any and/or which is more relevant cell line results, ex vivo or in vivo on healthy subjects, or infected patients or mouse model.  This valuable point of critical review makes the review more informative, enhancing understanding and raising points to be thought about for future plans of research to fill these gaps.

A clear example: page 2, 3 line 67  - 71

Ciporofloxacin decrease IL-1β from human monocytes in response to LPS ex-vivo. Then reported in another study (Ref. 16, Bailly et al. 1991) that ciprofloxacin increased proinflammatory cytokines including IL-1β.

Pasting from text: “In vivo production of proinflammatory cytokines in healthy volunteers in healthy volunteers is enhanced by ciprofloxacin”.

Comment, suggested to be in the text of the manuscript : one can conceivably say that the decrease in IL-1β in Bailly et al. 1990, (Ref. 15) is less relevant because the ciporofoxacin was added to the monocytes ex-vivo, whereas in Bailly et al. 1991, (Ref. 16)  Ciporofloxacin was given orally to healthy subjects for 7 consecutive days. This is more relevant to the in vivo effect, where the antibiotic is given orally to patients.  

Of Note, blood monocytes were challenged with LPS ex-vivo in both studies. The authors should mention LPS-stimulation in the text when citing from Ref. 16. The text should be reported in more informative way for the reader to understand and guide minds of the expected reason behind difference in IL-1β pattern. In both studies , monocytes were LPS-stimulated but the antibiotic (Ciporofloxacin) was studied in vivo for 7 days unlike the ex-vivo addition of Ciporofloxacin.

It is also conceivable to add to the text: “although Bailly et al. 1991 in their in vivo study of oral administration of Ciporofloxacin (Ref. 16) is more relevant, one limitation in that study is the missed timely interaction between LPS-challenge and administration of antibiotic, where LPS-challenge should be injected intraperitoneally in vivo followed by Ciporofloxacin oral ingestion 1 to 3 days later to better simulate clinical setting of septic shock. This is feasible but difficult in human and can be done in mouse models. Alternatively septic patients can be suitable subjects for this study. The effect of Ciporofloxacin on the inflammatory cytokines will be reliably concluded from future studies that will address the experimental limitations, mentioned above.

Another point of critical review: page 3, 4, line 120-123 under sub-title: 2.3. lymphocyte proliferation.

Since the current review focused on cellular immunity, and the sub-tile is lymphocyte proliferation, one can see those lines 120-123 citing published results on cancer cells. It is more coherent flowing structure to narrate results on apoptosis and proliferation from studies on patients’ cancer cells or cancer cell lines to be gathered in one separate section under sub-title “effect of antibiotics on apoptosis and proliferation of cancer cells”. A valid comment to be stated as the reviewers’ words in, among others, the text could be:

The studies showing inhibition of proliferation and/or enhanced apoptosis of cancer cells by antibiotic treatment, alone or combined with drug raise prospective beneficial effect of antibiotics in cancer treatment beyond the sole classic known anti-microbial effect of antibiotics. It is worthy to counter for the negative effect of antibiotics on immunity by giving these patients pro-biotic or dose of the beneficial bacteria families, known to be affected by the antibiotic(s)  of future clinical studies.

The phrase – focus on cellular immunity – conciliate/does not conflict with putting one sub-title on antibiotics effect on proliferation and apoptosis in cancer cells.

Author Response

The authors addressed an increasingly important negative side effects of antibiotics on immunity despite the beneficial effect in treatment and fighting infections, especially bacterial ones.

However, apart from the recognized and appreciated narration of the published studies with focus on the context of sepsis and the negative effect of antibiotics on cellular immunity, the manuscript is, in some parts, lacking the appropriate critical review on the published data and the clear complete drafting of the cited results  e.g., when the results of several studies on the same point are different or contradictory, the review drafting authors should critically discuss the possible reason of difference, if any and/or which is more relevant cell line results, ex vivo or in vivo on healthy subjects, or infected patients or mouse model. This valuable point of critical review makes the review more informative, enhancing understanding and raising points to be thought about for future plans of research to fill these gaps.

A clear example: page 2, 3 line 67 - 71

Ciporofloxacin decrease IL-1β from human monocytes in response to LPS ex-vivo. Then reported in another study (Ref. 16, Bailly et al. 1991) that ciprofloxacin increased proinflammatory cytokines including IL-1β.

Pasting from text: “In vivo production of proinflammatory cytokines in healthy volunteers in healthy volunteers is enhanced by ciprofloxacin”.

Comment, suggested to be in the text of the manuscript : one can conceivably say that the decrease in IL-1β in Bailly et al. 1990, (Ref. 15) is less relevant because the ciporofoxacin was added to the monocytes ex-vivo, whereas in Bailly et al. 1991, (Ref. 16) Ciporofloxacin was given orally to healthy subjects for 7 consecutive days. This is more relevant to the in vivo effect, where the antibiotic is given orally to patients.

Of Note, blood monocytes were challenged with LPS ex-vivo in both studies. The authors should mention LPS-stimulation in the text when citing from Ref. 16. The text should be reported in more informative way for the reader to understand and guide minds of the expected reason behind difference in IL-1β pattern. In both studies , monocytes were LPS-stimulated but the antibiotic (Ciporofloxacin) was studied in vivo for 7 days unlike the ex-vivo addition of Ciporofloxacin.

It is also conceivable to add to the text: “although Bailly et al. 1991 in their in vivo study of oral administration of Ciporofloxacin (Ref. 16) is more relevant, one limitation in that study is the missed timely interaction between LPS-challenge and administration of antibiotic, where LPS-challenge should be injected intraperitoneally in vivo followed by Ciporofloxacin oral ingestion 1 to 3 days later to better simulate clinical setting of septic shock. This is feasible but difficult in human and can be done in mouse models. Alternatively septic patients can be suitable subjects for this study. The effect of Ciporofloxacin on the inflammatory cytokines will be reliably concluded from future studies that will address the experimental limitations, mentioned above.

Thank you for suggestions, we have rephrased this section as follows: “For instance, ciprofloxacin decreases release of IL-1ß from human volunteer monocytes stimulated for 24hours in vitro with lipopolysaccharide (LPS),[15] however after administration of oral ciprofloxacin to healthy volunteers for 7 days, in vitro LPS stimulation enhanced IL-1ß production.[16] Previous work by our group has also demonstrated that ciprofloxacin and LPS in vitro stimulation suppresses pro-inflammatory cytokine release from volunteer and septic patient PBMCs after 24 hours but not via mitochondrial pathways.[5] Whether these conflicting findings are due to differences between in vivo or in vitro models, or due to the differences in duration of antibiotic administration remains to be elucidated. Either animal models of sepsis or cells isolated from septic patients administered ciprofloxacin could be used to support these findings.”

Additionally we have added additional sentences and paragraphs of critical review throughout the manuscript and other areas of conflict within the literature.

Another point of critical review: page 3, 4, line 120-123 under sub-title: 2.3. lymphocyte proliferation.

Since the current review focused on cellular immunity, and the sub-tile is lymphocyte proliferation, one can see those lines 120-123 citing published results on cancer cells. It is more coherent flowing structure to narrate results on apoptosis and proliferation from studies on patients’ cancer cells or cancer cell lines to be gathered in one separate section under sub-title “effect of antibiotics on apoptosis and proliferation of cancer cells”. A valid comment to be stated as the reviewers’ words in, among others, the text could be:

The studies showing inhibition of proliferation and/or enhanced apoptosis of cancer cells by antibiotic treatment, alone or combined with drug raise prospective beneficial effect of antibiotics in cancer treatment beyond the sole classic known anti-microbial effect of antibiotics. It is worthy to counter for the negative effect of antibiotics on immunity by giving these patients pro-biotic or dose of the beneficial bacteria families, known to be affected by the antibiotic(s) of future clinical studies.

The phrase – focus on cellular immunity – conciliate/does not conflict with putting one sub-title on antibiotics effect on proliferation and apoptosis in cancer cells.

We feel it is important to continue the focus on cellular immunity, we have therefore chosen to rephrase the paragraph to highlight that the cancer findings suggest a possible mechanism for the effect seen in lymphocytes. “Antibiotics have previously been demonstrated to inhibit cell proliferation. In volunteer PBMCs, fluroquinolones impair proliferation through alterations in IL-2 release and increased monocyte prostaglandin E2 release.[43, 44][45] Additional mechanistic work in breast and lung cancer cell lines suggest the effect could also be caused by damaging mitochondria and deactivating PI3K/Akt/mTOR (phosphatidylinositol 3-kinase/Ak strain transforming/mechanistic target of rapamycin) and MAPK/ERK (extracellular signal-regulated kinases) pathways.[39, 40]”

Reviewer 3 Report

Comments and Suggestions for Authors

The presented review article is interesting and covers an important topic - antibiotic-induced immunosuppression, which is gains significance more and more as the antibiotic prescription is increasing in out-patient and in-patient settings. 

The manuscript is written well and easy to understand and follow.

I would suggest to the authors to add one or two sentences to the end of the Introduction section and state the aim of the current manuscript (review).

Also, it would be nice if some abbreviations (like ETC, ROS, COPD, etc.) are given in full when mentioned for the first time.

Author Response

The presented review article is interesting and covers an important topic - antibiotic-induced immunosuppression, which is gains significance more and more as the antibiotic prescription is increasing in out-patient and in-patient settings. 

The manuscript is written well and easy to understand and follow.

I would suggest to the authors to add one or two sentences to the end of the Introduction section and state the aim of the current manuscript (review).

We have added the following sentence “In this review we describe the effects of antibiotics on effects on mitochondrial function and leukocyte functions including myeloid cell chemotaxis, phagocytosis, cytokine production, and antigen presentation; and lymphocyte apoptosis and proliferation. We report findings from different clinically relevant in vivo and in vitro models.”

Also, it would be nice if some abbreviations (like ETC, ROS, COPD, etc.) are given in full when mentioned for the first time

All abbreviations now spelt out